# The Effects of Acid and Water in the Formation of Anodic Alumina: DFT and Experiment Study

**DOI:** 10.3390/molecules28062427

**Published:** 2023-03-07

**Authors:** Zhengwei Zhang, Jin Kang, Xiaodong Li, Ping Li, Yali Du, Yufan Qin, Ningyi Li, Jiebin Li

**Affiliations:** 1High Value Fine Chemicals Research Center, Department of Chemistry & Chemical Engineering, Jinzhong University, Jinzhong 030619, China; 2Shaanxi Applied Physics and Chemistry Research Institute, Xi’an 710061, China

**Keywords:** anodic alumina, acid radical, water, DFT

## Abstract

The DFT method is employed to study the adsorption and reaction behaviors of HC_2_O_4_^−^, H_2_PO_4_^−^, HSO_4_^−^ and H_2_O on neutral and anodic aluminum slabs. With the exception of adsorption, the three acid radicals can successively take the two H atoms from the adsorbed H_2_O on the anodic aluminum slabs, which is the key step of the formation of anodic alumina. The dehydrogenation reaction is dominated by the Coulombic interaction of O and H, respectively belonging to acid radicals and the adsorbed H_2_O or OH, rather than by the interaction of electronic orbits located on the two kinds of atoms. The experiment of anodic polarization of aluminum verifies the calculation result well.

## 1. Introduction

The anodic alumina layer has been used in many fields, such as corrosion resistance [1], capacitors [2], biomaterials [3] and biosensors [4].The alumina film obtained through the anodic oxidation of aluminum in acidic electrolytes has been investigated for its wide range of uses.

Many factors influence the morphology and crystal structure of anodic alumina. The anodizing parameters including anodic potential [5,6], electrolyte [7,8] and temperature [6,9,10,11] have been investigated. The electrolyte component is the key factor affecting anodic alumina. Oxalic acid, phosphoric acid and sulfuric acid solutions are usually used to form the anodic alumina. When the anodic oxidation was performed in oxalic acid solutions [12,13,14], the concentrations were usually 0.3 M, and the temperature, voltage and time varied with different morphologies. The solutions containing phosphoric acid [15,16,17,18,19,20] were usually used when fabricating porous alumina. Sulfuric acid [21,22,23] was also used in the fabrication of porous alumina. In most cases, the combined application of the three acids was carried out in order to regulate the size and morphology of pores in the anodic alumina. Some other acids [24,25,26,27] are occasionally used, and their aims are also obtaining various porous anodic alumina. The morphology of alumina film closely relates to acid radicals in the electrolyte.

The factors influencing the formation of alumina morphology have been carefully studied in experiments, but the microscopic mechanism at the molecular level was rarely involved. Microscopic mechanism investigation involves the change of the Al–O bond and the composition of oxides. Su [28] studied the field-enhanced water dissociation at the growing oxide surface in order to explain the dissolution of alumina and the pore formation in the anodic alumina. This work solved one of problems raised 22 years ago by Thompson and coworkers [29]. Thompson and coworkers [29] also mentioned phosphorous and carbon contained in the anodic alumina formed in phosphate and oxalate solution, respectively. They pointed out that phosphorus extends over about two thirds of the film thickness. Garcia-Vergara and coworkers [30] also found the phosphorous signal in the anodic alumina film fabricated by anodizing in Na_2_HPO_4_ electrolyte. The phosphorous is present from the surface of the film to relative depths of in the range ~0.7 to ~0.8 of the film thickness. Thompson [31] obtained a similar result. A relatively pure anodic alumina contains 3–4 wt% phosphorous [29]. The changing mechanism of the Al-O bond and composition of oxides need to be further explored.

What are the roles of acid radicals and water during the formation of anodic alumina? Does the oxygen of alumina come from the acid radicals or water? How does the oxygen of alumina become separated from the molecule? The initial adsorption and interaction of acid radicals and water have a relationship with those questions. In the present work, the adsorption and interaction of electrolyte molecules will be studied using the DFT method.

## 2. Results

### 2.1. The Adsorption of Molecules

The small molecules of H_2_O, HC_2_O_4_^−^, H_2_PO_4_^−^ and HSO_4_^−^ were randomly placed on three aluminum slabs of (100), (110) and (111), see Appendix A. The three slabs are, respectively, given 0 or +1 charges to represent neutral or anodic conditions in the calculation. The adsorption energies (E_ads_) of H_2_O and acid radicals are listed in Table 1. Their adsorption states are shown in Appendix A.

The E_ads_ is calculated using the following equation:E_ads_ = E_slab-molecule_ − E_slab_ − E_molecule_(1)
where the E_slab-molecule_ is the energy of the aluminum slab adsorbed with the small molecule, the E_slab_ is the energy of the aluminum slab and the E_molecule_ is the energy of the small molecule. All of the adsorption energies of H_2_O and the three acid radicals are negative. The four small molecules can stably adsorb on the three kinds of aluminum faces under neutral or anodic conditions. The adsorption energies of the three acid radicals are bigger than that of H_2_O, and the adsorption of acid radicals is more stable. The H_2_O and the three acid radicals have bigger E_ads_; in other words, the four small molecules have more stable adsorption states when the aluminum faces are in the anodic condition.

The directly stable adsorption is the first behavior of the H_2_O and the three acid radicals on the three aluminum faces.

### 2.2. The Acid Radicals Replacing the Adsorbed H_2_O

Because the three acid radicals adsorb more stably than H_2_O, the adsorbed H_2_O may be replaced by acid radicals leaving the aluminum surface. The models of the acid radicals substituting the adsorbed H_2_O were built as shown as Appendix A. The acid radicals approach the adsorbed H_2_O from the side. The calculation results show that the adsorbed H_2_O really can be replaced by the acid radicals. These courses are represented by Appendix A, too. The replacing energy values (E_rep_) of the three replacing courses are listed in Table 2. The E_rep_ is calculated using the following equation:E_rep_ = E_rep after_ − E_slab-water_ − E_acid radical_(2)
where the E_rep after_ is the energy of the adsorbed H_2_O replaced by the acid radical, the E_slab-water_ is the energy of the aluminum slab with adsorbed H_2_O and the E_acid radical_ is the energy of the acid radical. The negative E_rep_ values mean that the replacing courses are feasible.

The three acid radicals replacing the adsorbed H_2_O is the second behavior of the four small molecules.

### 2.3. The Acid Radicals Bonding with the H of Adsorbed H_2_O

The acid radicals of HC_2_O_4_^−^, H_2_PO_4_^−^ and HSO_4_^−^ have the O atoms enriched with negative charge, so they have the ability to bond with the H of adsorbed H_2_O. In the initial models, the O atom of the acid radical is near to the H of adsorbed H_2_O on the neutral aluminum slab. After the calculation of geometry optimization, the acid radical does connect with the adsorbed H_2_O through O–H–O, as shown as Figure 1. The H–O bond of H_2_O is stretched to a longer length. The change in bonding energy (E_bon_) of the acid radical and adsorbed H_2_O is calculated using the following equation:E_bon_ = E_(slab-H_2_O-acid radical)_ − E_slab-H_2_O_ − E_acid radical_(3)
where the E_(slab-H_2_O-acid radical)_ is the energy of the acid radical connecting with the adsorbed H_2_O on the neutral aluminum slab, the E_slab-H_2_O_ is the energy of the neutral aluminum slab with adsorbed H_2_O and the E_acid radical_ is the energy of the acid radical. The E_bon_ values of the three acid radicals are listed in Table 3. The negative E_bon_ values indicate that the connection between the acid radicals and adsorbed H_2_O are reasonable.

The three acid radicals connecting with the adsorbed H_2_O on the neutral aluminum slab is the third behavior of the four small molecules.

The adsorption of radicals is more stable than water, yet the molecular structures of radicals remain stable on the neutral or positively charged slab, see Appendix A. It has been confirmed that the concentrations of light elements are very low in the anodic alumina [32]. This implies that the adsorption of radicals with complete structure has no important contribution to Al–O formation. The O–H of adsorbed water is easily stretched by the radicals, see Figure 1 and Table 3, so the third behavior of water and radicals may be the dominant way in which the Al–O bond is formed.

### 2.4. The Acid Radicals Stripping off H from Adsorbed H_2_O and OH

On the neutral aluminum slabs, the acid radicals can stretch the H–O bond of adsorbed H_2_O to a longer length, but they do not have the ability to snap it. Under the anodic condition, the three acid radicals have enough power to successfully take the two H atoms away from the adsorbed H_2_O, see Figure 2. The O atom of the adsorbed H_2_O is finally left to bond with the aluminum on the surface. The processes of acid radicals stripping off H are represented through chemical equations in Figure 3. Every equation has two reaction directions. The “1” direction is the acid radical stripping off H from the adsorbed H_2_O or OH, and the “2” direction is the corresponding reverse reaction. The energy change of the reaction of the acid radical stripping off H (E_str_) is calculated using the following equation:E_str_ = E_(OH or O+acid)_ − E_slab-H_2_O or OH_ − E_acid radical_(4)
where the E_(OH or O+acid)_ is the energy of the acid and adsorbed OH (one of the H atoms of H_2_O is stripped) or adsorbed O (the two H atoms of H_2_O are stripped), the E_slab-H_2_O or OH_ is the energy of the anodic aluminum slab adsorbed with H_2_O or OH and the E_acid radical_ is the energy of the acid radical. Every step of the acid radical taking the H atom away from the adsorbed H_2_O or OH has a negative E_str_ value (see Table 4), in other words, the reverse reactions are not feasible.

The acid radical continuously stripping the two H atoms off the adsorbed H_2_O on the anodic aluminum slab is the fourth behavior of the four molecules. The binding energies of absorption for the three acid radicals (see Table 1) are actually lower than those of reacting with H_2_O (see Table 4). The reaction of extracting H from H_2_O has a larger tendency than adsorption for the three radicals. It can be deduced that the adsorption of radicals really occurs, with much less influence on the formation of the Al–O bond. In the course of alumina formation, the function of the acid radical is to pull off the two H atoms of the adsorbed H_2_O, and the role of H_2_O is the supplier of oxygen in alumina.

### 2.5. The Reasons for Acid Radicals Stripping off H Atoms of H_2_O

The acid radicals stripping off the H atoms of the adsorbed H_2_O are chemical reactions including the changing of chemical bonds. DFT energies about bond changes are some of the most accurate theoretical results, and can be interpreted by analysis of the Klopman–Salem equation [32] from a molecular orbital theory perspective. The Coulombic and/or covalent interactions between the acid radicals and H atoms determine the reactions in the Klopman–Salem equation:(5)$$\Delta E=\frac{Q_{\mathrm{Nu}}Q_{\mathrm{EI}}}{\varepsilon R}+\frac{2{\left({\beta C}_{\mathrm{Nu}}C_{\mathrm{EI}}\right)}^{2}}{E_{\mathrm{Nu},\mathrm{HOMO}}-{\;E}_{\mathrm{EI},\mathrm{HOMO}}}$$
where ΔE is the energy change of the reaction, Q_Nu_ is the charge of the nucleophile, Q_EI_ is the charge of the electrophile, ε is the local dielectric constant, R is the distance between the nucleophile and electrophile, β is the resonance integral, c is the coefficient of the molecular orbit to form the new bond, E_Nu,HOMO_ is the HOMO energy of the nucleophile and E_EI,LUMO_ is the LUMO energy of the electrophile. In the present work, the nucleophiles are acid radicals and the electrophiles are the H atoms of adsorbed H_2_O. In the Klopman– Salem equation, the first term and the second term represent the Coulombic interaction and orbital interaction between acid radicals and H atoms, respectively.

If the orbital interaction determines the reaction of acid radicals stripping off H atoms, the ΔE should inversely relate to the difference between E_Nu,HOMO_ and E_EI,LUMO_, as shown in the second term of the Klopman–Salem equation. Figure 4 shows the difference between E_acid radical,HOMO_ (the HOMO energy of the acid radical) and E_H_ (the orbital energy of H in the adsorbed H_2_O and OH). The value of (E_HSO4_^−^_,HOMO_ − E_H_) is the smallest one, which is close to that of (E_H2PO4_^−^_,HOMO_ − E_H_). The value of (E_HCO4_^−^_,HOMO_ − E_H_) is the largest one. In Table 4, the reaction energy change of HSO_4_^−^ stripping off H is the lowest one, the H_2_PO_4_^−^ has the largest reaction energy change and the HCO_4_^−^ has a large reaction energy change, too. The results in Table 4 have no regular relationship with the values of (E_acid radical,HOMO_ − E_H_), and, thus, the orbital interaction does not determine the reaction of acid radicals stripping off H atoms.

If the Coulombic interaction determines the reaction of acid radicals stripping off H atoms, the ΔE should positively relate to the product of Q_Nu_Q_EI_, as shown in the first term of the Klopman–Salem equation. Figure 5 shows the charges of the O atom of HCO_4_^−^, H_2_PO_4_^−^ and HSO_4_^−^. These O atoms directly interact with the H atoms with charge of Q_H_. The order of charge values is Q_O, H2PO4_^−^ > Q_O, HCO4_^−^ > Q_O, HSO4_^−^, so there should be Q_H_Q_O, H2PO4_^−^ > Q_H_Q_O, HCO4_^−^ > Q_H_Q_O, HSO4_^−^ for the three acid radicals interacting with the same kind of H. In Table 4, the reaction energy changes of HCO_4_^−^, H_2_PO_4_^−^ and HSO_4_^−^ are also in the order of H_2_PO_4_^−^ > HCO_4_^−^ > HSO_4_^−^. The order of ΔE is same as that of the products of Q_H_Q_O, acid radical_, so ΔE positively relates to the product of Q_H_Q_O_. The reactions of dehydrogenation may be dominated by the Coulombic interaction.

Since the reaction is solely affected by Coulombic interaction (the first term of the Klopman–Salem equation), there are two further inferences. The first one is that the acid radical with larger-charged O has the greater ability to extract the same kind of H atom of adsorbed H_2_O. The second one is that it is easier for a larger-charged H of adsorbed H_2_O to be stripped off by the same acid radical.

The positive charge of the slab makes the charge of H of adsorbed H_2_O and OH increase, as shown as Table 5. This means the Q_H, neutral_ is smaller than the Q_H, positive_. The product of Q_H, neutral_Q_O, acid radical_ is smaller than that of Q_H, positive_Q_O, acid radical_. The product value of charges indicates that the reaction of stripping off H on the positively charged slab is easier. The H atoms of H_2_O adsorbed on the neutral slab really cannot be stripped off, see Figure 1, while those on the positively charged slab can be taken away by the same acid radical, see Figure 2. The second inference is reasonable.

According to the courses of proving the two inferences, the acid radical with larger-charged O in favor of the dehydrogenation of adsorbed H_2_O and the positive charge of the anodic slab are necessary for accelerating the process of the H leaving the adsorbed H_2_O.

### 2.6. The Experimental Verification

The calculation results, see Table 4, indicate that the abilities of dehydrogenation of the three acids are in the order phosphoric acid > oxalic acid > sulfuric acid. In other words, the forming of the anodic alumina is the easiest in the phosphoric acid aqueous solution, and it is the most difficult in the sulfuric acid aqueous solution.

In order to verify the theoretic results, the anodic alumina will, respectively, grow in the three acid solutions. The aluminum foil with (100), (110) and (111) faces, see Figure 6a, is immersed into the acid solution to perform anodic polarization. In the sulfuric acid solution, the current rises with the increasing potential of the aluminum anode. In the oxalic acid solution, the change in current has a similar trend to that in the sulfuric acid solution, but the current of the oxalic acid solution is smaller than that of the sulfuric acid solution. In the phosphoric acid solution, the current maintains the smallest value, and the polarization curve is far under the curves generated in the oxalic and sulfuric acids solutions, as shown in Figure 6b. The reduction in the current arises from the coverage of the anodic surface by the growth of alumina. The polarization curves of Figure 6b imply that the phosphoric acid has the most powerful ability to extract H from the H_2_O adsorbed on the anodic surface, and to facilitate the growth of anodic alumina. The sulfuric acid is the weakest one. The oxalic acid lies between the former two acids.

The experiment result has the same trend as the theoretic calculation. It verifies the theoretic interpretation of the formation course of anodic alumina.

The acid radical stepwisely removes the hydrogen atoms of H_2_O adsorbed on the anodic surface. The ability of dehydrogenation is in the order H_3_PO_4_ > H_2_CO_4_ > H_2_SO_4_. The reaction of dehydrogenation is dominated by the Coulombic interaction between the oxygen anion of the acid radical and the hydrogen of the water, rather than by the frontier molecular orbits of the acid radical and water.

This work only supplies a reference for selecting acid and interprets the initial stage of alumina formation. The incrassation of alumina involves the transfer of oxygen and aluminum ions in the alumina layer and the dehydrogenation of water on the alumina surface. These processes need a follow-up study.

## 3. Materials and Methods

### 3.1. Computation

The oxalic acid, phosphoric acid and sulfuric acid are frequently used as the acid component of electrolyte in preparing anodic alumina. In the aqueous solution of oxalic acid, phosphoric acid or sulfuric acid, the three kinds of acid radicals HC_2_O_4_^−^, H_2_PO_4_^−^ or HSO_4_^−^, respectively, are the dominant species. The (100), (110) and (111) are the main exposed faces on the aluminum surface. The water, three acid radicals and Al (100), (110) and (111) faces were selected as the specific model to investigate the roles of water and acid in the formation of the Al–O bond of anodic alumina.

Al (100), (110) and (111) slabs (p5 × 5) with 5 layers and 20 Å vacuum range were built, see Appendix A. The atomic coordinates of the inner three layers (blue) are fixed, while the atoms of the outer two layers (pink) relax freely. The small molecules including H_2_O, HC_2_O_4_^−^, H_2_PO_4_^−^ and HSO_4_^−^ were placed on the center of slabs in the course of calculation. Computations were performed using DMOl3 code, which adopts fully self-consistent DFT calculations to solve Kohn–Sham equations. The generalized gradient approximation (GGA), with the functional PBE for metallic surfaces adsorbing with some small molecules [33], was employed. For all models, the double numerical plus polarization (DNP) [34] was selected as the basis set. The ultrasoft pseudopotentials for Al, S, P, O, C and H were used in all calculations. The energy convergence tolerance was 2 × 10^−5^ eV per atom.

### 3.2. Experiment

Commercial aluminum foils was prepared. They simultaneously have three faces of (100), (110) and (111), which was confirmed through the analysis of X-ray diffraction (XRD) on a PAN-analytical-X’Pert PRO X diffractometer equipped with Cu Kα radiation (λ = 0.15406 nm). The aluminum foils were cut into small squares with a tail as the outgoing line. The side length of the squares is 10 mm. The squares were washed subsequently with acetone and ethanol. Electrochemical tests were successively performed in the oxalic acid solution (0.1 mol/L), phosphoric acid solution (0.1 mol/L) and sulfuric acid solution (0.1 mol/L). The Versatile Multichannel Potentiostat 2/Z (VMP2, Princeton Applied Research) was employed in the test. The Ag/AgCl reference electrode was adopted to obtain the relative potential. The potential sweeping speed is 10 mV s^−1^ in this work.

## 4. Conclusions

The H_2_O, HCO_4_^−^, H_2_PO_4_^−^ and HSO_4_^−^ have four kinds of adsorption behaviors on the aluminum slabs: (1) all of them can spontaneously adsorb on the aluminum slabs; (2) the HCO_4_^−^, H_2_PO_4_^−^ and HSO_4_^−^ have the ability to replace the adsorbed H_2_O; (3) the HCO_4_^−^, H_2_PO_4_^−^ and HSO_4_^−^ can connect with the adsorbed H_2_O through the O–H–O bond on the neutral aluminum slabs and (4) on the anodic aluminum slabs, the HCO_4_^−^, H_2_PO_4_^−^ and HSO_4_^−^ connected with the adsorbed H_2_O can further polarize the O–H bond of H_2_O and strip off its H. The three radicals can continuously extract the second H of H_2_O. The process going on in cycles leaves the O being left to form alumina. The third and fourth behaviors are the key steps of the formation of anodic alumina. The effect of radicals is to take the H away from the H_2_O. The effect of the H_2_O is to supply the O for anodic alumina.

The dehydrogenation processes between the acid radicals and adsorbed H_2_O on the anodic aluminum slab are thermodynamically feasible and have no potential barriers. These processes are dominated by the Coulombic interaction of the O of radicals and H of adsorbed H_2_O or OH, rather than by the interaction of electronic orbits located on the two kinds of atoms. The experiment verifies the theoretic interpretation of formation course of anodic alumina well.

The energy changes of extracting H on the three faces are different. The effect of aluminum slab structures influence the anodic alumina formation, too. This factor will be investigated in the follow-up study.

## Figures and Tables

**Figure 1 molecules-28-02427-f001:**
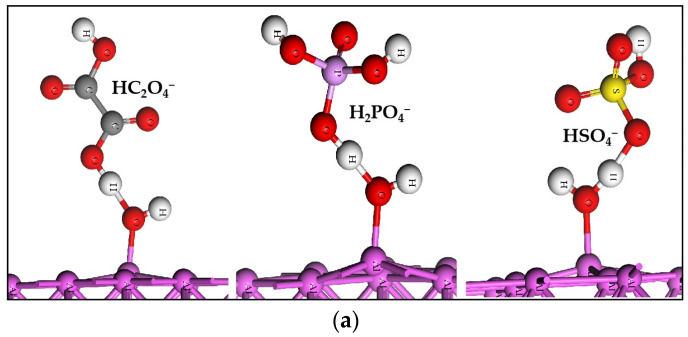
The acid radicals connecting with the adsorbed H_2_O through O–H–O bond: (**a**) the (100), (**b**) the (110), (**c**) the (111) surfaces.

**Figure 2 molecules-28-02427-f002:**
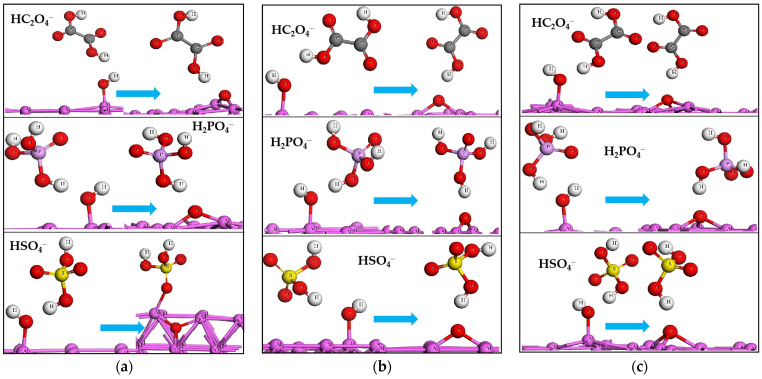
The acid radicals stripping off the two H of H_2_O by two steps on the anodic aluminum slabs: (**a**) the (100)^+1^, (**b**) the (110)^+1^, (**c**) the (111)^+1^ surfaces.

**Figure 3 molecules-28-02427-f003:**
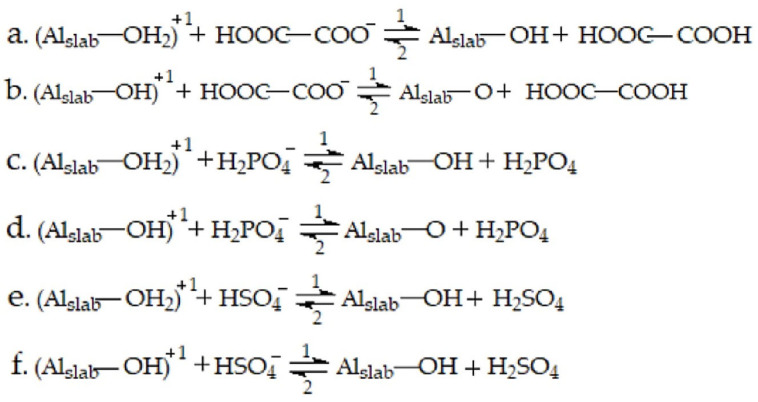
The equations of acid radicals stripping off H in H_2_O on the anodic aluminum slabs.

**Figure 4 molecules-28-02427-f004:**
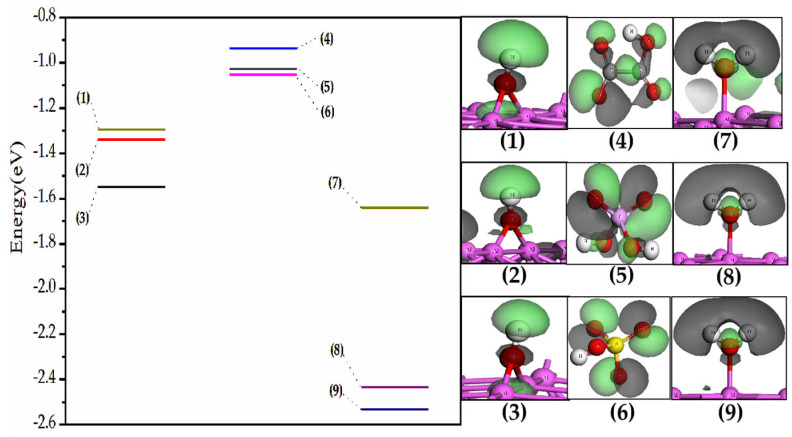
The energy levels and shapes of H in OH adsorbing on the anodic aluminum slabs: (**1**) (111)^+1^, (**2**) (110)^+1^, (**3**) (100)^+1^; the energy levels and shapes of HOMO of acid radicals: (**4**) HCO_4_^−^, (**5**) H_2_PO_4_^−^, (**6**) HSO_4_^−^; the energy levels and shapes of H in H_2_O adsorbing on the anodic aluminum slabs: (**7**) (110)^+1^, (**8**) (100)^+1^, (**9**) (111)^+1^.

**Figure 5 molecules-28-02427-f005:**
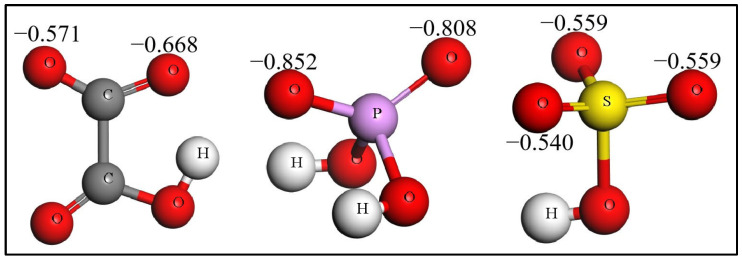
The charges of the O atoms in acid radicals.

**Figure 6 molecules-28-02427-f006:**
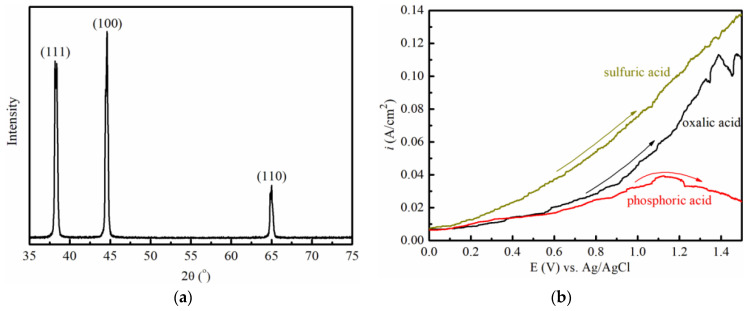
(**a**) The XRD peaks of aluminum foil. (**b**) The anodic polarization of aluminum in acid solutions.

**Table 1 molecules-28-02427-t001:** The adsorption energies (E_ads_/eV) of small molecules.

Slabs	H_2_O	HC_2_O_4_^−^	H_2_PO_4_^−^	HSO_4_^−^
(100)^0^	−0.34	−3.65	−4.06	−2.93
(110)^0^	−0.47	−3.39	−3.28	−3.08
(111)^0^	−0.36	−2.78	−3.70	−3.18
(100)^+1^	−0.50	−4.62	−5.16	−3.69
(110)^+1^	−0.53	−3.90	−3.67	−3.75
(111)^+1^	−0.56	−3.99	−5.22	−3.80

**Table 2 molecules-28-02427-t002:** The E_rep_ (eV) values of acid radicals replacing H_2_O.

Slabs	HC_2_O_4_^−^	H_2_PO_4_^−^	HSO_4_^−^
(100)^0^	−3.31	−3.72	−2.59
(110)^0^	−2.92	−2.81	−2.61
(111)^0^	−2.42	−3.33	−2.82
(100)^+1^	−4.11	−4.65	−3.18
(110)^+1^	−3.37	−3.14	−3.22
(111)^+1^	−3.42	−4.65	−3.23

**Table 3 molecules-28-02427-t003:** The E_bon_ (eV) values of acid radicals bonding with the H of adsorbed H_2_O.

Slabs	HC_2_O_4_^−^	H_2_PO_4_^−^	HSO_4_^−^
(100)^0^	−2.23	−1.41	−0.52
(110)^0^	−1.70	−0.83	−0.04
(111)^0^	−2.49	−1.73	−0.80

**Table 4 molecules-28-02427-t004:** The E_str_ (eV) values of acid radicals stripping off one of the H in H_2_O. The a, b, c, d, e and f correspond to the labels of reactions in Figure 3. Subscript “1” is for the direction of reactions in Figure 3.

Slabs	a_1_	b_1_	a_1_ + b_1_	c_1_	d_1_	c_1_ + d_1_	e_1_	f_1_	e_1_ + f_1_
(100)^+1^	−3.21	−2.54	−5.75	−3.24	−2.57	−5.81	−2.57	−1.91	−4.48
(110)^+1^	−2.33	−2.15	−4.48	−2.42	−2.24	−4.66	−1.76	−1.58	−3.34
(111)^+1^	−2.97	−3.37	−6.34	−3.08	−3.48	−6.56	−2.49	−2.89	−5.38

**Table 5 molecules-28-02427-t005:** The charges of the two H of adsorbed H_2_O; the charges of the H of adsorbed OH.

Slabs	H	H	Slabs	H
(100)^0^-H_2_O	0.274	0.274	(100)^0^-OH	0.261
(110)^0^-H_2_O	0.267	0.267	(110)^0^-OH	0.260
(111)^0^-H_2_O	0.282	0.282	(111)^0^-OH	0.298
(100)^+1^-H_2_O	0.322	0.322	(100)^+1^-OH	0.323
(110)^+1^-H_2_O	0.311	0.311	(110)^+1^-OH	0.312
(111)^+1^-H_2_O	0.328	0.328	(111)^+1^-OH	0.343

## Data Availability

Data generated in this study are stored at Jinzhong University and are available upon reasonable request.

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
