# Peer review of "The Effects of Acid and Water in the Formation of Anodic Alumina: DFT and Experiment Study"

_molecules, 2023, doi:10.3390/molecules28062427_

Round 1

Reviewer 1 Report

The manuscript is well-written and clearly presented by the authors.

The content seems to be original and scientifically robust, the authors tried to deliver the mechanistic route of adsorption phenomena of different acids and water on the neutral and anodic aluminum slabs by means of experimental and theoretical investigations

I recommend this for publication in this Journal in the present form.

The only query is about the subheading need to be followed by the journal's pattern

Author Response

Answer : Thanks for the reviewer’s friendly reminder, the subheading is reedited.

Reviewer 2 Report

In this manuscript, Zhang et al. investigated the mechanism of anodic alumina formation in the environment of acid solutions. The authors built theoretical models of aluminum slabs and water molecules, in complex with three major types of acid radicals respectively. Some key properties regarding the formation of anodic alumina, such as adsorption, replacement/binding to H2O, as well as stripping off hydrogen from H2O, were assessed using density functional theory (DFT)-calculated energies. The theoretical results indicate that phosphoric acid is the best among all the acids studied in this work to help form anodic alumina, which was supported by electrochemical polarization experiments.

The main scope of this manuscript is suitable for Molecules. However, a large part of the results lacks in-depth interpretation and discussion to convey meaningful conclusions to lead to the main theme of this work. Some existing discussions (detailed below) do not follow rigorous reasoning and strongly confuse readers. As a result, the scientific significance and quality of presentation of this manuscript are very limited, and I don’t recommend its acceptance on Molecules. My major concerns are elaborated on below.

1.   The first three parts of the results as described in the current manuscript – adsorption (2.1), acid replacing H2O (2.2), and acid bonding with H of adsorbed H2O (2.3) – do not directly correlate with the major topic of this manuscript – the formation of anodic alumina. The data in these three parts need to be accompanied by a more reasonable discussion following clear logic to address why these behaviors are significant for the formation of anodic alumina.

2.    In 2.4, the authors calculated the energy for different acids to strip the hydrogen of water near aluminum slabs by DFT methods. Then in 2.5, they attempted to explain the scientific reasoning of results from a physical/chemical perspective by describing different components of the Klopman-Salem equation. This reasoning is extremely vague because the authors assume the DFT-calculated energies are “correct” ground truth and weight components in the Klopman-Salem equation in a purely qualitative way. It would be more scientifically clear and sound to calculate the energy using the Klopman-Salem equation, compare them with the DFT results, and discuss their consistency/discrepancy. 

3.    This study modeled three types of aluminum slabs named (100), (110), and (111). However, the authors did not define or show their full atomic structures. The hydrogen-stripping energies in Table 4 – key results leading to the major conclusion of this study – clearly depend on the types of aluminum slabs. The effect of aluminum slab structures could be important for anodic alumina formation and requires further explanation and discussion.

4.    The English writing of this manuscript has many serious issues to hinder understanding of its content, including but not limited to:

- inappropriate assertive words (e.g. “The second inference is correct”)

- redundant words/sentences (e.g. “The first inference is proved to be true by …”)

- grammar errors (especially incorrect usage of commas)

- typos (too many to list)

This manuscript needs extensive English editing/rewriting to reach the level of a qualified scientific publication. 

Author Response

Reviewer 2.

  1. 1.   The first three parts of the results as described in the current manuscript – adsorption (2.1), acid replacing H2O (2.2), and acid bonding with H of adsorbed H2O (2.3) – do not directly correlate with the major topic of this manuscript-the formation of anodic alumina. The data in these three parts need to be accompanied by a more reasonable discussion following clear logic to address why these behaviors are significant for the formation of anodic alumina.

Answer: The correlation of adsorption of radicals and water with formation of alumina is interpreted in the third and the fourth paragraphs of introduction and the the last paragraph of 2.3. The changed words is red.

  1. In 2.4, the authors calculated the energy for different acids to strip the hydrogen of water near aluminum slabs by DFT methods. Then in 2.5, they attempted to explain the scientific reasoning of results from a physical/chemical perspective by describing different components of the Klopman-Salem equation. This reasoning is extremely vague because the authors assume the DFT-calculated energies are “correct” ground truth and weight components in the Klopman-Salem equation in a purely qualitative way. It would be more scientifically clear and sound to calculate the energy using the Klopman-Salem equation, compare them with the DFT results, and discuss their consistency/discrepancy. 

Answer: In 2.5, the primary purpose is to clear up the decisive factor of radicals extracting of H from H2O and OH. The values of ΔE in Tab.4 are the reality. We have to explain those values by looking for a proper theory. The decisive factor of radicals extracting of H is found out by the explanation of those energy values. This is the logical basis for writing the 2.5 part. We think the discussion sequence of 2.5 may be better on the original text. The part of 2.5 is not reedited.

  1. This study modeled three types of aluminum slabs named (100), (110), and (111). However, the authors did not define or show their full atomic structures. The hydrogen-stripping energies in Table 4 – key results leading to the major conclusion of this study – clearly depend on the types of aluminum slabs. The effect of aluminum slab structures could be important for anodic alumina formation and requires further explanation and discussion.

Answer: We have noticed the influence of different face, too. The follow-up study would focus on the influence of different face by using of foil with single face. Our group study anode film of aluminium electrolytic capacitor. We usually use the aluminium foil with single (100) face which is also used at manufacturing plant. The follow-up study would explore the alumina film growing on different single face. In the present paper the effects of acid radicals and water were emphasis.

  1. The English writing of this manuscript has many serious issues to hinder understanding of its content, including but not limited to:

- inappropriate assertive words (e.g. “The second inference is correct”)

- redundant words/sentences (e.g. “The first inference is proved to be true by …”)- grammar errors (especially incorrect usage of commas)

- typos (too many to list)

Answer: Those issues mentioned by reviewer are revised.

Reviewer 3 Report

Zhang et al. have used experimental and computational methods to ascertain the impact of water and acid radicals on the synthesis of alumina anodic alumina. Studies indicate that the manuscript is appropriate for publication. Prior to making an editorial decision, the following considerations must be made:

1.       A detailed introduction is required.

2.       Authors can make their manuscripts easier to read by rearranging the figures and tables, for example, if authors reference the primary figure first, then the supporting figures, that will be beneficial.

3.       Figure 4 is unclear, has terminology mentioned explicitly, and must be rearranged.

4.       The energy terms for water and acid radicals have been estimated by the authors, who have then contrasted the deprotonation or hydrogen abstraction of hydrogen (water) in the presence of acid radicals. If these radicals are offered in a competitive manner, what will happen?

5.       The study came to the conclusion that the water and acid radical adsorption was caused by coulombic interactions. What causes phosphoric acid to have a higher charge on oxygen than sulfuric and oxalic acid?

6.       For metallic surfaces, the generalized gradient approximation (GGA) using the functional PBE was used. The decision needs to be supported. The information on packages utilized and techniques employed in this work must be included in the computational details. Authors have not gone into detail on the creation of the computational surface.

Author Response

Reviewer 3.

  1. A detailed introduction is required.

Answer: The some detailed references are added into the introduction. The changed words is red.

  1. Authors can make their manuscripts easier to read by rearranging the figures and tables, for example, if authors reference the primary figure first, then the supporting figures, that will be beneficial.

Answer: Thanks for the reviewer’s friendly reminder, the figures and tables are rearranged.

3.Figure 4 is unclear, has terminology mentioned explicitly, and must be rearranged.

Answer: Thanks for the reviewer’s friendly reminder, the figure 4 is rearranged.

4.The energy terms for water and acid radicals have been estimated by the authors, who have then contrasted the deprotonation or hydrogen abstraction of hydrogen (water) in the presence of acid radicals. If these radicals are offered in a competitive manner, what will happen?

Answer: If the “radicals are offered in a competitive manner” means mixed radicals?  Different radical simultaneously extracts the H in the adsorbed OH or H2O in the same solution or on the same surface. Their rates of extracting hydrogen are different. The rate is mainly influenced by the Gibbs free energy change of extracting hydrogen reaction. The rate difference influence the morphology of alumina, perhaps, so the mixed radicals were used in the forming of porous alumina.

  1. The study came to the conclusion that the water and acid radical adsorption was caused by coulombic interactions. What causes phosphoric acid to have a higher charge on oxygen than sulfuric and oxalic acid?

Answer: The electronegativities of O, P, C and S elements orderly are 3.44, 2.19, 2.55 and 2.58. The degree of polarization of O-P, O-C, O-S bonds decrease gradually. The atoms in the highly polarized bond have more charges. This leads the charges of O element in H2PO4-, HC2O4- and HSO4- to decrease in turn, so phosphoric acid has a higher charge on oxygen than sulfuric and oxalic acid.

  1. For metallic surfaces, the generalized gradient approximation (GGA) using the functional PBE was used. The decision needs to be supported. The information on packages utilized and techniques employed in this work must be included in the computational details. Authors have not gone into detail on the creation of the computational surface.

Answer: The GGA-PBE method is adopted widely. The supporting reference [36] is supplemented. The three surface modes of aluminum are supplemented in the supplement materials, see Fig. S1. The construction method of modes are described in 3.1.

Round 2

Reviewer 2 Report

The edits in the revised version improved the quality of this manuscript. I recommend its publication after these corresponding revisions have been made:

My concern #1 in round 1 review:

The new discussions in the introduction help readers understand the scope of investigating the first three parts. However, in the last paragraph of 2.3, the authors still fail to explain the fundamental reason why “the adsorption of radicals with the complete structure has no important contribution to Al-O formation”. The binding energies of absorption for three acid radicals are actually lower than bonding with H2O. More rigorous discussion is needed here.

My concern #2 in round 1 review:

DFT indeed yields more accurate energy values than the molecular orbital theory-based Klopman-Salem equation. But it is still a theory instead of “reality” (this kind of assertive word should be avoided in scientific publications unless under extreme conditions). At the beginning of 2.5, the authors should make a clear point that DFT energies are one of the most accurate theoretical results and they would like to interpret the data by analysis of the Klopman-Salem equation from a molecular orbital theory perspective.

My concern #3 in round 1 review:

This concern has been adequately addressed.

My concern #4 in round 1 review:

Some English issues were fixed. But there are still too many grammar errors (e.g. “…The morphology of alumina film is closely relates to acid radicals in electrolyte…”), that significantly disrupt reading and understanding of the contents. 

Author Response

review2-1:

The new discussions in the introduction help readers understand the scope of investigating the first three parts. However, in the last paragraph of 2.3, the authors still fail to explain the fundamental reason why “the adsorption of radicals with the complete structure has no important contribution to Al-O formation”. The binding energies of absorption for three acid radicals are actually lower than bonding with H2O. More rigorous discussion is needed here. 

Answer: Thanks a lot for further pointing out the flaw. We have discussed the reason of “the adsorption of radicals with the complete structure has no important contribution to Al-O formation” according to the reviewer’s hint. The discussion is added into the last paragraph of 2.4.

 review2-2:

DFT indeed yields more accurate energy values than the molecular orbital theory-based Klopman-Salem equation. But it is still a theory instead of “reality” (this kind of assertive word should be avoided in scientific publications unless under extreme conditions). At the beginning of 2.5, the authors should make a clear point that DFT energies are one of the most accurate theoretical results and they would like to interpret the data by analysis of the Klopman-Salem equation from a molecular orbital theory perspective.

Answer: we would try to avoid some assertive words. At the beginning of 2.5, the clear point has been added according to reviewer’s hint.

review2-3:

This concern has been adequately addressed.

review2-4:

Some English issues were fixed. But there are still too many grammar errors (e.g. “…The morphology of alumina film is closely relates to acid radicals in electrolyte…”), that significantly disrupt reading and understanding of the contents. 

Answer: we have modified the paper in detail. Some grammar errors were put right.